# Biosynthesis of Phenylamide Phytoalexins in Pathogen-Infected Barley

**DOI:** 10.3390/ijms20225541

**Published:** 2019-11-06

**Authors:** Naoki Ube, Yukinori Yabuta, Takuji Tohnooka, Kotomi Ueno, Shin Taketa, Atsushi Ishihara

**Affiliations:** 1United Graduate School of Agriculture, Tottori University, Tottori 680-8553, Japan; nube.tottori@gmail.com; 2Faculty of Agriculture, Tottori University, Tottori 680-8553, Japan; yabuta@tottori-u.ac.jp (Y.Y.); kotoueno@tottori-u.ac.jp (K.U.); 3National Agriculture and Food Research Organization, Tsukuba 305-8518, Japan; tohnooka@affrc.go.jp; 4Institute of Plant Science and Resources, Okayama University, Kurashiki 710-0046, Japan; staketa@rib.okayama-u.ac.jp

**Keywords:** *Hordeum vulgare*, phytoalexin, phenylamide, triticamide, tryptamine *N*-hydroxycinnamoyl transferase, *Bipolaris sorokiniana*, *Fusarium culmorum*, *Fusarium graminearum*

## Abstract

Phytoalexins are inducible antimicrobial metabolites in plants, and have been indicated to be important for the rejection of microbial infection. HPLC analysis detected the induced accumulation of three compounds **1**–**3** in barley (*Hordeum vulgare*) roots infected by *Fusarium culmorum*, the causal agent of Fusarium root rot. Compounds **1**–**3** were identified as cinnamic acid amides of 9-hydroxy-8-oxotryptamine, 8-oxotryptamine, and (1*H*-indol-3-yl)methylamine, respectively, by spectroscopic analysis. Compounds **1** and **2** had been previously reported from wheat, whereas **3** was an undescribed compound. We named **1**–**3** as triticamides A–C, respectively, because they were isolated from barley and wheat, which belong to the Triticeae tribe. These compounds showed antimicrobial activities, indicating that triticamides function as phytoalexins in barley. The administration of deuterium-labeled *N*-cinnamoyl tryptamine (CinTry) to barley roots resulted in the effective incorporation of CinTry into **1** and **2**, which suggested that they were synthesized through the oxidation of CinTry. Nine putative tryptamine hydroxycinnamoyl transferase (THT)-encoding genes (*HvTHT1*–*HvTHT9*) were identified by database search on the basis of homology to known THT gene sequences from rice. Since *HvTHT7* and *HvTHT8* had the same sequences except one base, we measured their expression levels in total by RT-qPCR. *HvTHT7*/*8* were markedly upregulated in response to infection by *F. culmorum*. The HvTHT7 and HvTHT8 enzymes preferred cinnamoyl- and feruloyl-CoAs as acyl donors and tryptamine as an acyl acceptor, and (1*H*-indol-3-yl)methylamine was also accepted as an acyl acceptor. These findings suggested that HvTHT7/8 are responsible for the induced accumulation of triticamides in barley.

## 1. Introduction

Plants possess a variety of defense mechanisms for preventing pathogen infection, including the accumulation of specialized metabolites. In response to pathogen infection, many plant species accumulate antimicrobial metabolites, i.e., phytoalexins, which provide chemical defense [1]. The importance of chemical defense in plants has been indicated by various kinds of evidence: the lack of phytoalexin resulted in the increased susceptibility of plants [2], and accumulation of phytoalexin in other plants by introduction of biosynthetic genes enhanced resistance levels [3]. In addition, some pathogens have been found to counteract to phytoalexin production by acquisition of the degradation and excretion mechanism of phytoalexins [4].

Phenylamides are amides of hydroxycinnamic and benzoic acids, with various biogenic amines, and have been reported to contribute to plant defense responses. Representative hydroxycinnamic acids components of phenylamides include cinnamic (Cin), *p*-coumaric (Cou), caffeic (Caf), ferulic (Fer), and benzoic (Ben) acids, whereas representative amine components include tryptamine (Try), serotonin (Ser), tyramine (Tyr), agmatine (Agm), and putrescine (Put). Hereafter, specific phenylamides will be referred to using combined abbreviations of their constituent acids and amines, e.g., CinTry (cinnamoyltryptamine). Phenylamide accumulation has been reported to occur in a wide range of plant species. In the Poaceae species, various phenylamides have been reported to accumulate in response to pathogen attack and have been indicated to function as phytoalexins in plant chemical defenses. In rice (*Oryza sativa*), for example, plants have been reported to accumulate BenTry and CinTry, as phytoalexins, in response to attack by rice blast fungus (*Pyricularia oryzae*) [5] and to accumulate FerTry, CouSer, FerSer, and Ser in response to infection by brown spot fungus (*Cochliobolus miyabeanus*) [6]. In addition, rice leaves have even been reported to accumulate phenylamides (CinTry, CinTyr, CouSer, and BenTry) in response to ultraviolet (UV) light exposure [7,8]. Meanwhile, in oats, avenanthramides, which are hydroxycinnamic acid amides with anthranilic acids, function as phytoalexins [9] and as substrates for the reinforcement of cell walls [10]. Furthermore, wheat (*Triticum aestivum*) leaves have recently been reported to accumulate *N*-cinnamoyl-8-oxotryptamine and *N*-cinnamoyl-9-hydorxy-8-oxotryptamine, as well as CouAgm, FerAgm, CouPut, FerPut, and FerSer in response to attack by *Bipolaris sorokiniana*, which is the causal agent of brown spot [11], and it is possible that *N*-cinnamoyl-8-oxotryptamine and *N*-cinnamoyl-9-hydorxy-8-oxotryptamine function as phytoalexins, on the basis of their anti-microbial properties.

Phenylamides are biosynthesized via the condensation of hydroxycinnamoyl-CoA esters and amines, and this reaction is catalyzed by a variety of hydroxycinnamoyl transferases [12,13]. In rice, for example, tryptamine benzoyl transferase (TBT), tryptamine hydroxycinnamoyl transferase (THT), and putrescine hydroxycinnamoyl transferase (PHT) contribute to the biosynthesis of the phenylamides that are constitutively present in rice leaves [14,15], and in oats, hydroxyanthranilate *N*-hydroxycinnamoyl transferase (HHT) is involved in the biosynthesis of avenanthramide phytoalexins [16,17]. The enzymes involved in the synthesis of these phenylamides are classified into the BAHD acyltransferase family, which was named for the first letter of each of the first four biochemically characterized enzymes in the family, namely BEAT (benzoyl alcohol *O*-acetyltransferase), AHCT (anthocyanin *O*-hydroxycinnamoyl transferase), HCBT (anthranilate *N*-hydroxycinnamoyl/benzoyl transferase), and DAT (deacetyl vindoline 4-*O*-acetyltransferase) [12]. In barley, the role of agmatine *N*-coumaroyltransferase (ACT), which is also included in the BAHD acyltransferase family, in the biosynthetic of CouAgm has been well studied [18,19]. However, the enzymes and genes involved in the synthesis of phenylamides from aromatic amines have not been investigated, despite that tyramine hydroxycinnamoyl transferase activity has been detected in young seedling roots [20].

The accumulation of specialized metabolites has been investigated in pathogen-challenged barley plants. *Bipolaris sorokiniana*-infected barley leaves, for example, were reported to accumulate the dimer of serotonin and 3-(2-aminoethyl)-3-hydroxyindolin-2-one, which may function as phytoalexins [21], and *Fusarium graminearum*-infected barley roots were reported to exude cinnamic, *p*-coumaric, ferulic, syringic, and vanillic acids [22]. Furthermore, the metabolomics analysis performed by Karre et al. [23] revealed differences in the accumulation of metabolites in the spikelets of *F. graminearum*-resistant and -susceptible cultivars, and the authors reported that phenolics, lignans, hydroxycinnamic acid amides (phenylamides), flavonoids, signaling molecule-related compounds, terpenoids, indole alkaloids, and methionine biosynthesis-related compounds contributed to *Fusarium* head blight resistance that was mediated by chitin elicitor receptor kinase. Even more recently, barley leaves have been reported to accumulate cinnamoyl-9-hydroxy-8-oxotryptamine and cinnamoyl-8-oxotryptamine, which function as phytoalexins in wheat, in response to CuCl_2_ treatment [11]. In barley, phenylamides that contain Agm and Put are reportedly induced by pathogen infection [23,24]. However, no other inducible phenylamides have been characterized, despite the characterization of multiple phenylamides from the poaceous species. Accordingly, the aim of the present study was to investigate the inducible phenylamides of barley, in the hopes of identifying an undiscovered chemical defense mechanism. As a result, three phenylamide phytoalexins, including a previously undescribed compound, were identified, and acyltransferase genes that encode the enzymes that catalyze inducible phenylamide synthesis were characterized. On the basis of these findings, the present study provides novel insight into the chemical defense mechanisms of barley, as well as a framework for the biosynthesis of phenylamide phytoalexins.

## 2. Results

### 2.1. Detection of Inducible Metabolites

The causal agent of barley root rot, *F. culmorum*, was cultured on V8 agar medium, and plugs of the agar medium were used to inoculate 4-d-old barley seedlings. Metabolites were extracted from the inoculated roots using 80% methanol 72 h after inoculation, and the extracts were subjected to HPLC analysis. As shown in Figure 1, three compounds (**1**–**3**) were observed to accumulate in *F. culmorum*-inoculated leaves, and after HPLC purification, **1**–**3** were subjected to LC-MS analysis. Compounds **1** and **2** exhibited [M+H]^+^ ions at *m/z* 321 and *m/z* 305, respectively, and comparison of the detected ions and the retention times with those of reference compounds identified the compounds as *N*-cinnamoyl-9-hydorxy-8-oxotryptamine and *N*-cinnamoyl-8-oxotryptamine, respectively (Figure 2). The accumulation of these compounds has also been reported to be induced in barley leaves by treatment with 1 mM CuCl_2_ [11]. Compounds **1** and **2** were designated as triticamides A and B, respectively, because they were first isolated from wheat and barley, both of which belong to the Triticeae tribe in the Poaceae family. Compound **3** exhibited [M+H]^+^ ions at *m/z* 277. However, this ion did not correspond with any inducible compounds that had been previously reported from barley.

### 2.2. Identification of **1**–**3**

Compound **3** was purified from 1.5 kg *F. culmorum*-inoculated barley roots and was determined to possess a molecular formula of C_18_H_16_ON_2_ using HRMS (*m/z* 277.1331, [M+H]^+^). The ^1^H-NMR and COSY spectra of **3** (Table 1) indicated two groups of aromatic proton signals at δ_H_ 7.00, 7.10, 7.35, and 7.59 ppm for a 1,2-disubstituted benzene ring and at δ_H_ 7.35–7.41 and 7.54 ppm for a 1-monosubstituted benzene ring. In addition, the ^1^H-NMR spectrum also revealed the presence of two coupled double-bond protons (δ_H_ 6.69 and 7.49 ppm, *J* = 15.6 Hz), a methylene proton (δ_H_ 4.56 ppm), and three 1H protons (δ_H_ 7.31, 8.35, and 10.95 ppm). Furthermore, the ^1^H- and ^13^C-NMR spectra of **3** were similar to those of **2** in the aromatic region (Table 1). Comparison of the ^13^C-NMR spectra of **2** and **3** indicated an upfield shift of the methylene carbon at δ_C_ 45.9 ppm in **2** to δ_C_ 34.1 ppm in **3**, and the disappearance of a carbonyl carbon signal in **2** (δ_C_ 190.1 ppm, C-8). These differences corresponded to differences in the molecular formulas of **3** (C_18_H_16_ON_2_) and **2** (C_19_H_16_O_2_N_2_). Together, these results suggested that **3** is the cinnamic acid amide of (1*H*-indol-3-yl)methylamine (IMA). To confirm this, the compound was synthesized using the condensation of *N*-hydroxysuccinimide ester of cinnamic acid and IMA. The reaction resulted in the formation of *N*-cinnamoyl-(1*H*-indol-3-yl)methylamine with a yield of 86.4%. The NMR spectra of the synthetic cinnamic acid amide of IMA was identical to those of **3**. Thus, **3** was unequivocally identified to be *N*-cinnamoyl-(1*H*-indol-3-yl)methylamine. Because **3** has not been previously described, it was designated triticamide C.

### 2.3. Accumulation of 1–3 after Pathogen Infection

The accumulation kinetics of **1**–**3** were investigated in *F. culmorum-*infected roots (Figure 3A). The amounts of **1**–**3** in barley roots increased from 24 h after inoculation, reached maximum levels at 72 h after inoculation, and decreased thereafter. In addition, none of the three compounds were detected in control roots. The inoculation of roots with *F. graminearum* and *B. sorokiniana* also induced the accumulation of **1**–**3**, and at 72 h after inoculation, the amounts of **1**–**3** reached 86.7, 2.8, and 1.3 nmol/g FW, respectively, in the *F. graminearum*-infected roots and 90.8, 5.2, and 1.1 nmol/g FW in the *B. sorokiniana*-infected roots (Figure 3B). To investigate the effect of pathogen infection on the accumulation of **1**–**3** in barley leaves, a suspension of *B. sorokiniana* conidia was inoculated onto the third leaves of three-week-old seedlings (Figure 3C). At 72 h after inoculation, **1**–**3** reached 31.4, 10.3, and 7.0 nmol/g FW, respectively, in the *B. sorokiniana*-infected leaves. In addition, none of the three compounds were detected in control or intact leaves. To investigate the generality of the accumulation of **1**–**3** in barley, *F. culmorum* was inoculated onto three different cultivars ‘Yumesakiboshi’, ‘CDC Fibar’, and ‘Morex’. Accumulation of **1**–**3** was induced in the infected roots of the all cultivars 72 h after inoculation (Figure 3D). These findings indicated that the accumulation of **1**–**3** is a general response of barley both in roots and leaves, irrespective to the pathogen species.

The amounts of tryptamine, 8-oxotryptamine, and IMA in *F. culmorum*-infected barley roots were also measured by LC-MS/MS, using multiple reaction monitoring (MRM) methods, since the compounds could be precursors of **1**–**3**. Tryptamine was detected at 0.5 nmol/g FW in control roots, whereas neither 8-oxotryptamine nor IMA were detectable, and levels of all three increased following the inoculation of roots with *F. culmorum* (Figure 3E), with the amounts of tryptamine, 8-oxotryptamine, and IMA reaching 24.4, 8.7, and 39.9 nmol/g FW, respectively. Thus, the production of amines is a part of inducible response against pathogen infection.

Even though the accumulation of **1** and **2** had been previously observed in pathogen-infected wheat leaves [11], their accumulation in wheat roots has yet to be investigated. Therefore, the accumulation of **1**–**3** was examined by HPLC analysis of extracts from *F. culmorum*-infected wheat roots. Compounds **1** and **2** were observed to accumulate in the pathogen-infected roots, to levels of 353 and 2.0 nmol/g FW, respectively, whereas **3** was not detected (Appendix A).

### 2.4. Accumulation of Various Phenylamides in Barley Roots and Leaves in Response to Pathogen Infection

Because **1**–**3** were classified as phenylamides on the basis of their chemical structures, the accumulation of other phenylamides in *F. culmorum*-infected roots and *B. sorokiniana*-infected leaves was investigated. A total of 25 phenylamide combinations of five acids (Cin, Cou, Caf, Fer, and Ben) and five amines (Try, Ser, Tyr, Agm, and Put) were measured simultaneously using LC-MS/MS as previously described [25]. The induced accumulation of CinPut, FerPut, CinAgm, CinTyr, CouTry, FerTry, CouPut, and FerSer was observed in the *F. culmorum*-infected roots. However, only CouPut and FerSer exhibited marked increases (Figure 4). Furthermore, even though CouAgm levels were not increased in response to infection by *F. culmorum*, the concentrations were consistently high (86.6 and 83.3 nmol/g FW in control and inoculated roots, respectively).

Meanwhile, the induced accumulation of FerPut, CouAgm, FerTyr, and FerSer was observed in *B. sorokiniana*-infected leaves (Figure 5), and CouAgm exhibited the greatest accumulation, followed by FerSer, FerPut, and FerTyr. Accumulation was also observed for CinPut, CouPut, CinAgm, CouTyr, CinTry, and FerTry; however, their concentrations were relatively low. The phenylamides detected in the *F. culmorum*-infected roots were similar to those observed in *B. sorokiniana*-infected leaves. CinTyr and CouTry exhibited a small but significant accumulation in the *F. culmorum*-infected roots, whereas CouTyr exhibited the significant accumulation in the *B. sorokiniana*-infected leaves.

These findings indicated that the biosynthetic pathways leading to multiple phenylamides are simultaneously activated in the defense response to pathogen infection.

### 2.5. Induced Expression of the Tryptamine Cinnamoyl Transferase-Like Genes in Pathogen Infected Barley

In rice, tryptamine hydroxycinnamoyltransferases (OsTHT1/2) and tryptamine benzoyltransferases (OsTBT1/2) have been reported to contribute to the synthesis of CouSer and BenTry, respectively [14]. Therefore, it is plausible that their barley homologs could be involved in the biosynthesis of triticamides. Indeed, BLAST search of the barley genome sequences using rice THT1/2 and TBT1/2 protein sequences as queries detected nine homologous genes (named *HvTHT1*–*HvTHT9*, Appendix A). Phylogenetic analysis of the amino acid sequences grouped HvTHT1–HvTHT9 into a clade IVb that also contained OsTHTs and OsTBTs in the BAHD acyltransferase family (Figure 6A).

Gene expression of *HvTHT1–HvTHT9* in roots and leaves at 24 h after inoculation with pathogenic fungi was investigated by qRT-PCR. Because there is only a single base difference in the nucleotide sequences of *HvTHT7* and *HvTHT8* at 495 bp from the start codon, their transcripts could not be distinguished by PCR. Therefore, the combined transcript levels of *HvTHT7* and *HvTHT8* were measured using a single primer set for both genes. The total expression of *HvTHT7/8* in the *F. culmorum*-infected roots was 60 times that of control roots, whereas the total expression of *HvTHT7*/*8* in *B. sorokiniana*-infected leaves was 206 times that of control leaves (Figure 6B). In addition, *HvTHT2* and *HvTHT5* were upregulated by 27- and 10-fold in *B. sorokiniana*-infected leaves, but their enhanced expression was not detected in *F. culmorum*-infected roots. In regard to accumulation kinetics, the *HvTHT7*/*8* transcripts attained a maximum 24 h after inoculation and then rapidly decreased thereafter (Figure 6C). The expression of other *HvTHT* genes were also monitored in the *F. culmorum* infected roots. The incresed accumulation of transcrpts of *HvTHT1*, *HvTHT3*, *HvTHT4*, *HvTHT6*, and *HvTHT9* was detected 48 h after inocualtion, but their leves were at most 6 times that of control roots (Appendix A). Based on these findings, the *HvTHT7/8* were the genes that respond to pathogen infection most sharply.

To examine if both *HvTHT7* and *HvTHT8* were expressed in response to pathogen attack, the fragments of *HvTHT7* and *HvTHT8* containing the single base substituted site at 495 bps from the start codon were amplified by PCR from the cDNA prepared from *F. culmorum*-infected root of seedlings of the cultivar ‘Shunrei’. The DNA sequencing of the fragments showed only the fragment from *HvTHT8* was amplified, indicating that the *HvTHT8* was mainly expressed in the *F. culmorum*-infected root (Appendix A).

We also performed database search of the *HvTHT7* and *HvTHT8* in the genome sequences of ‘Barke’ and ‘Bowman’ (URL: https://webblast.ipk-gatersleben.de/barley_ibsc/). In the database of ‘Barke’ genome, only *HvTHT7* sequences were found whereas, in the database of ‘Bowman’ genome, only sequences of *HvTHT8* were detected. We also performed the PCR to amplify the fragments containing the single base substituted site using the genomic DNAs prepared from ‘Shunrei’ and ‘Morex’, and then sequenced the amplified fragments. We only detected the sequences corresponding to *HvTHT8 in* ‘Shunrei’ but both sequences of *HvTHT7* and *HvTHT8* in ‘Morex’, indicating that ‘Shunrei’ poses only *HvTHT8* whereas ‘Morex’ poses both *HvTHT7* and *HvTHT8*.

### 2.6. Characterization of Barley THTs

To characterize the enzymatic functions of HvTHT7 and HvTHT8, the proteins tagged with His-GST were heterologously expressed in *Escherichia coli* BL21. The *HvTHT7* cDNA was generated from *HvTHT8* cDNA using site-directed mutagenesis, and the recombinant proteins were detected in soluble protein fractions of *E. coli* lysate using CBB stain (Appendix A). The His-GST-tagged enzymes were purified to homogeneity using metal-affinity chromatography, and the acyltransferase activities of the recombinant enzymes were assayed using various CoA thioesters of cinnamic acid relatives as acyl donors and biogenic amines as acyl acceptors. The amounts of reaction products were determined using LC-MS/MS analyses with MRM methods.

Kinetic analysis revealed that HvTHT7 and HvTHT8 possessed similar enzymatic properties, which is not surprising when considering that amino acid sequences of the two proteins differ by only a single substitution (histidine to arginine at 166 aa). Both enzymes functioned optimally at 35 °C and pH 8.0. In the presence of 1 mM tryptamine as an acyl acceptor, the relative efficacy of the enzymes did not differ largely among the CoA esters (Table 2). The values for the Cin- and Fer-CoAs were nearly identical, and the value for Cou-CoA was the approximately half that for Cin-CoA. In the presence of 200 μM Cin-CoA as an acyl donor, the *K*_m_ values for tryptamine were considerably lower than those for 8-oxotryptamine, serotonin, and IMA, and the *k*_cat_ values for tryptamine were higher than those for 8-oxotryptamine, IMA, and serotonin. Accordingly, the relative efficiencies of the enzymes for tryptamine were much larger than those for other tested amines. These findings demonstrated that both HvTHT7 and HvTHT8 favor Cin- and Fer-CoAs over Cou-CoA as acyl donors and favor tryptamine over other tryptamine derivatives as an acyl acceptor, which suggests that HvTHT7 and HvTHT8 are involved in the biosynthesis of CinTry in barley.

For comparison, recombinant HvTHT2 was also produced as described above for HvTHT7 and HvTHT8. In the presence of tryptamine as an acyl acceptor, HvTHT2 strongly favored Fer-CoA over Cou- and Cin-CoAs as an acyl donor (Table 3). For example, the relative efficiency of CinTry formation was only 0.03% that of FerTry formation. The relative efficiencies for tryptamine and serotonin formation were similar, but that for tyramine formation was 35.6% that for tryptamine. On the basis of this substrate specificity, HvTHT2 is likely involved in the synthesis of FerTry an FerSer.

### 2.7. Feeding Experiments with Cinnamic Acid Amides in Pathogen-Infected Barley Roots

To confirm the pathway for the synthesis of **1** and **2** from cinnamic acid amides, [^2^H_5_]-CinTry was fed to *F. culmorum*-infected roots 48 h after inoculation. The roots were incubated with the labeled compound for 24 h and then extracted using 80% methanol, and the resulting extracts were subjected to LC-MS/MS analyses with MRM methods. [^2^H_5_] -CinTry was effectively incorporated into **1** and **2** at rates of 10.3% and 42.3%, respectively (Figure 7). *Fusarium culmorum*-infected roots were also fed [^2^H_5_]-**2**. However, the labeled compound was only incorporated into 2.87% of products, likely owing to the low water solubility of [^2^H_5_]-**2**. These findings indicated that the pathway from CinTry to **1** via **2** is functional.

### 2.8. Antimicrobial Activities of Triticamide C (**3**)

The antimicrobial activities of triticamides A and B have been reported previously. Here, the antifungal activities of triticamide C against *F. culmorum*, *F. graminearum,* and *B. sorokiniana* were assessed using inhibition assays for conidial germination and germ tube elongation. Triticamide C inhibited the conidial germination of *F. culmorum*, *F. graminearum*, and *B. sorokiniana* at concentrations of >100 μM (Figure 8A). However, complete inhibition was not observed even at 1000 μM. Furthermore, triticamide C also inhibited the conidial germ tube elongation of *F. graminearum* and *F. culmorum* even at 30 µM, and almost completely inhibited those of *F. graminearum* and *F. culmorum* at 1000 μM. The inhibitory rate for *B. sorokiniana* were 48.1% even at 1000 µM (Figure 8B). Meanwhile, in regards to antibacterial activity, triticamide C inhibited the growth of *P. syringae* pv. *japonica* at 100 μM but failed to yield complete inhibition even at 300 μM (Figure 8C). Triticamide C was an antimicrobial compound as well as triticamides A and B.

## 3. Discussion

The present study demonstrated that barley roots accumulate three inducible compounds, namely *N*-cinnamoyl-9-hydroxy-8-oxotryptamine (**1**) and *N*-cinnamoyl-8-oxotryptamine (**2**), and *N*-cinnamoyl-(1*H*-indol-3-yl)methylamine (**3**), in response to pathogen attack (Figure 2). Compounds **1** and **2** had already been characterized as phytoalexins in wheat [11], but compound **3** was a previously undescribed compound. Because both barley and wheat belong to the Triticeae tribe, the compounds **1**–**3** were named triticamides A–C, respectively. The antifungal activities of triticamides A and B have been reported previously [11], and the present study demonstrated that triticamide C also possesses antifungal activity against several species of phytopathogenic fungi (*F. culmorum*, *F. graminearum*, and *B. sorokiniana*). It is of interest to note that triticamide C inhibited the growth of the bacterial pathogen *P. syringae* at 100 µM because triticamides A and B did not affect the growth of *P. syringae* [11]. Triticamides function as phytoalexins in barley as previously demonstrated in wheat.

The concentrations of triticamides A–C were 282, 56.7 and 55.0 nmol/g FW, respectively, in *F. culmorum*-infected roots 72 h after inoculation, and 31.4, 10.3, and 7.0 nmol/g FW, respectively, in *B. sorokiniana-*infected leaves 48 h after inoculation (Figure 3). These concentrations are similar to the concentrations at which the triticamides exert antifungal activities. For example, triticamides B and C significantly inhibited the germ tube elongation of *F. graminearum* at 10 µM, and triticamides B and C inhibited the germination of *B. sorokiniana* conidia at 10 and 100 µM, respectively (Figure 8). The accumulation of triticamides likely affects the growth of these pathogens in plant tissues.

Phenylamides are biosynthesized by the condensation of hydroxycinnamic acid CoA thioesters and amines. Among the THT- and TBT-related genes that were identified in the barley genome, the combined expression levels of *HvTHT7*/*8* were markedly higher in pathogen-infected roots and leaves and reached maximum levels 24 h after inoculation, which preceded the accumulation of triticamides (Figure 3). Generally speaking, the accumulation of inducible metabolites is preceded by enhanced gene expression. For instance, in rice leaves, the amount of rice phytoalexin sakuranetin reached at a maximum 72 h after the start of treatment whereas the transcript amount of *NOMT* that encodes naringenin-7-*O*-methyltransfearase reached at a maximum 6 h after the treatment [26]. Similarly, the accumulation of avenanthramides in oat leaves was about 24 h behind the expression of biosynthetic genes [17]. To explain this gap of timings of *HvTHT7/8* expression and triticamide accumulation, however, the kinetic analysis of expression of other biosynthetic genes is needed. Some of remaining *HvTHT* genes showed enhanced expression in *F. culmorum*-infected root, but the fold changes were relatively small (Appendix A). In addition, the characterization of heterologously expressed HvTHT7 and HvTHT8 revealed that both enzymes favored Cin-CoA as acyl donors and tryptamine as an acyl acceptor (Table 2). Together, these findings suggest that HvTHT7/8 are involved in triticamide biosynthesis.

The genomic sequences of *HvTHT7* and *HvTHT8* were different only at one nucleotide at 497 bps from the start codon. In the present study, we mainly used the cultivar ‘Shunrei’. In this cultivar, we detected only the transcript of *HvTHT8* and the genomic sequence corresponding to *HvTHT8.* Thus, ‘Shunrei’ probably poses only *HvTHT8.* By contrast, we detected the transcripts and genomic sequences corresponding to both *HvTHT7* and *HvTHT8* in ‘Morex’. This is consistent with the result of database search–both *HvTHT7* and *HvTHT8* were deposited to ‘Morex’ genome database (URL: https://webblast.ipk-gatersleben.de/barley_ibsc/). The ‘Morex’ probably poses both genes in its genome. Furthermore, only *HvTHT7* was deposited in ‘Barke’ genome database, whereas only *HvTHT8* was in ‘Bowman’ genome database (URL: https://webblast.ipk-gatersleben.de/barley_ibsc/). Thus, the barley genome is polymorphic at this specific nucleotide in *HvTHT7/8* but some cultivars such as ‘Morex’ acquired both *HvTHT7* and *HvTHT8* during the breeding process. Because the enzymatic function of HvTHT7 and HvTHT8 were not largely different, this single nucleotide polymorphism does not affect the triticamide production.

There are two hypothetical pathways for the biosynthesis of **1** and **2**: the direct condensation of 9-hydroxy-8-oxotryptamine and 8-oxotryptamine with Cin-CoA thioester or the condensation of tryptamine and Cin-CoA into CinTry and the subsequent oxidation of the tryptamine part of the compound. The high substrate specificity of HvTHT7 and HvTHT8 for tryptamine suggests that the second hypothesis is more likely (Table 2). In addition, observed accumulation of tryptamine was much greater than that of 8-oxotryptamine (Figure 3). Indeed, subsequent feeding experiments confirmed that labeled CinTry was effectively incorporated into triticamides A and B (Figure 7). Thus, it is most likely that **1** and **2** are biosynthesized by this route. However, 8-oxotryptamine was also detected in the pathogen-infected roots and HvTHT7 and HvTHT8 accepted 8-oxotryptamine to some extent. Thus, the possibility that triticamides are synthesized by the direct condensation should not be excluded, and the pathways may form a metabolic grid. To obtain a whole picture of the pathways for triticamide biosynthesis, the concentrations and kinetics of 9-hydroxy-8-oxotryptamine in barley roots should be investigated.

Meanwhile, HvTHT2 preferred Fer-CoA as an acyl donor and tryptamine and serotonin as acyl acceptors. The *k*_cat_ value of HvTHT2 for Cin-CoA was much smaller than that for Fer-CoA, and the relative efficiency for Cin-CoA was only 0.033% that for Fer-CoA (Table 3). Therefore, it is unlikely that HvTHT2 contributes substantially to the biosynthesis of triticamides. The expression of *HvTHT2* was enhanced by *B. sorokiniana* infection in leaves but not by *F. culmorum* infection in roots, and both FerTyr and FerSer were observed to accumulate to high levels in *B. sorokiniana*-infected leaves (Figure 4, Figure 5 and Figure 6). Because HvTHT2 accepted tyramine and serotonin as an acyl acceptor, it is likely that *HvTHT2* is involved in the formation of such amides, rather than that of triticamides.

Triticamide C is considered to be biosynthesized by the condensation of Cin-CoA and IMA. Indeed, the recombinant enzymes HvTHT7 and HvTHT8 catalyzed this condensation reaction efficiently, and IMA accumulation was induced by *F. culmorum* infection to 39.9 nmol/g FW in infected roots (Table 2 and Figure 3). Therefore, the present study confirms the hypothesis that triticamide C is formed by the direct condensation of Cin-CoA and IMA. This amine is also a precursor of gramine, a specialized metabolite that is also present in barley [27]. Gramine is formed by the successive methylation of IMA by a methyltransferase [28]. However, the biosynthetic pathway from tryptophan to IMA has yet to be fully elucidated. Interestingly, several barley cultivars (e.g., ‘Morex’) do not accumulate gramine [28]. Therefore, it is worth noting that, in the present study, all the examined cultivars including ‘Morex’ accumulated triticamide C and, thus, produce IMA in response to pathogen infection.

The BAHD acyltransferase family is separated into five clades (Clades I-V), and Clade IV is further divided into Clades IVa and IVb (Figure 6) [12,14]. In the present study, HvTHT2, HvTHT7, and HvTHT8 were mapped to Clade IVb along with OsTHTs and OsTBTs. The catalytic activities of enzymes in this subfamily have been studied in several grass species, and all the studied enzymes have been reported to catalyze the condensation of tryptamine with either Ben-CoA or Cou-CoA [14]. In the present study, HvTHT2, HvTHT7, and HvTHT8 also accepted tryptamine as a substrate. Thus, it is reasonable to infer that enzymes in this subclade similarly prefer indole amines as substrates. Peng et al. [14] also identified Clade IV- (EVDSWL and VLWAFP) and Clade IVb-specific (VRVAVNC and RRRR) motifs in enzymes from rice (Appendix A). These motifs were also present in HvTHT2, HvTHT7, and HvTHT8, with the exception of the RRRR motif. Analyzing the functions of these motifs may help elucidate the structural factors that affect substrate specificity.

On the other hand, HvTHT7 and HvTHT8 showed different substrate specificity for CoA thioesters in comparison with OsTHTs and OsTBTs. HvTHT7 and HvTHT8 preferred Fer-CoA as well as Cin-CoA while OsTBTs and OsTHT1 showed low specificity to Fer-CoA although specificity of OsTHT2 for Fer-CoA has not been examined. OsTBTs and OsTHTs showed high specificities for Cou- and Ben-CoAs. We could obtain the sequences of 115 BAHD family acyltransferases by BLAST search using HvTHT7 amino acid sequences as query from the species in Poaceae family (Appendix A). Phylogenetic analysis of the acyltransferases indicated that HvTHT7 and HvTHT8 formed a cluster together with THTs in *T. aestivum*, *T. urartu*, *T. durum, Aegilops tauschii, Secale cereale, and Brachypodium distachion*. All of these species belong to the Pooideae subfamily. Thus, the acyltransferases in this cluster may have branched off from the cluster containing HvTHT2, OsTHT1/2, and OsTBT1/2, and play a specific role in Pooideae species. In this context, it is of interest to investigate the distribution of triticamide phytoalexins in these species as well as to analyze the substrate specificity of THTs in this cluster.

## 4. Materials and Methods

### 4.1. General Experimental Procedures

Both ^1^H and ^13^C NMR spectra and 2D COSY, HMQC, and HMBC spectra were recorded using an Avance II instrument (Bruker, Billerica, MA, USA). High-resolution mass spectra were measured using an Exactive mass spectrometer (Thermo Fisher Scientific, Waltham, MA, USA), and ESI-MS measurements were performed using a Quattro Micro API mass spectrometer (Waters, Milford, MA, USA) that was connected to an Acquity UPLC system (Waters). HPLC was performed using a 10A HPLC system (Shimadzu, Kyoto, Japan).

### 4.2. Plant Materials and Pathogenic Fungi

Barley (*Hordeum vulgare* ‘Shunrei’) and wheat (*Triticum aestivum* ‘Norin 61′) seeds were purchased from JA Inaba Tottori (Tottori, Japan) and Tsurushin Shubyo (Kyoto, Japan), respectively. Seeds of barley ‘Yumesakiboshi’, ‘CDC fibar’ and ‘Morex’ were stocks in the Institute of Plant Science and Resources, Okayama University, Japan. For the experiments that involved roots, seeds were placed on filter paper in a plant culture container (ф120 mm diammeter × 80 mm hight, SPL Life Science, Gyeonggi-do, Korea) that contained 5 mL sterile water, incubated at 4 °C in the dark for 24 h, and then incubated at 25 °C with a 14-h photoperiod for 3 days. For the experiments that involved leaves, seeds were immersed in distilled water for one night, sown on a 1:1 (*v*:*v*) mixture of vermiculite (Shoei Sangyo, Okayama, Japan) and culture soil (Iris Ohyama, Sendai, Japan), and then incubated at 25 °C with a 14-h photoperiod for three weeks.

*Fusarium culmorum* (MAFF 236454), the causal agent of *Fusarium* root rot, was obtained from the National Institute of Agrobiological Sciences Genebank (http://www.gene.affrc.go.jp/index_en.php), whereas *F. graminearum* was obtained from the National Agriculture and Food Research Organization, and *B. sorokiniana* (OB-25-1), the causal agent of barley spot blotch, was a stock in Natural Product Chemistry Laboratory, Faculty of Agriculture, Tottori University. The *F. culmorum*, *F*. *graminearum,* and *B*. *sorokiniana* were inoculated onto V8 agar plates. To inoculate plant roots, the *F. culmorum, F. graminearum*, and *B*. *sorokiniana* were cultured for 5 d at 25 °C under black light (FL15BL-B; Hitachi, Tokyo, Japan), and plugs from the V8 agar plates were used as inoculum. Meanwhile, conidia suspensions were used to inoculate plant leaves. To obtain conidia, *F. culmorum* and *F. graminearum* were cultured for one week under the conditions described above, whereas *B*. *sorokiniana* was cultured for two weeks.

The causal agent of bacterial black node, *Pseudomonas syringae* pv. *japonica* (MAFF 301072), was obtained from the National Institute of Agrobiological Sciences Genebank. The pathogen was cultured on potato semi-synthetic medium (300 g potato, 2 g Na_2_PO_4_·12H_2_O, 15 g sucrose, 5 g peptone, 0.5 g Ca(NO_3_)_2_·4H_2_O, and 1 L distilled water) at 25 °C with shaking (200 rpm) for 24 h.

### 4.3. Inoculation of Pathogenic Fungi

The inoculation of pathogens onto barley and wheat roots was performed according to the methods of Covarelli et al. [29] with slight modification. Briefly, mycelial agar plugs (0.5- to 0.6-cm diameter) were taken from the growing edges of a 5-d-old culture plates and placed on the roots of the 96-h-old barley seedlings at 1.5 cm away from the seed. The culture dishes that contained the barley seedlings were then incubated at 25 °C with a 14-h photoperiod for 24–120 h. The inoculated roots were extracted using 10 volumes of 80% methanol, and the extracts were analyzed by HPLC, as follows: column, Cosmosil 5C_18_-AR-II 4.6 I.D. × 150 mm (Nacalai Tesque, Kyoto, Japan); gradient, 5–70% B/(A+B) within 30 min, A: 0.1% trifluoroacetic acid aq., B: acetonitrile; flow rate, 0.8 mL/min; column temperature, 40 °C; detection, 280 nm. Compounds **1**, **2**, and **3** were eluted at 24.2, 25.5, and 26.5 min, respectively.

To inoculate barley and wheat leaves, droplets (5 μL) of the *B. sorokiniana* conidial suspension (5 × 10^5^ conidia/mL) were placed on the leaves of 3-week-old seedlings at 1.0-cm intervals. The inoculated seedlings were then kept in a moist air-tight bag for 24 h, removed from the bag and further incubated at 25 °C with a 14-h photoperiod for 48 h, and then extracted using 10 volumes of 80% methanol. The extracts were analyzed by HPLC.

### 4.4. Purification and Identification of **3**

The roots (1.5 kg) of barley ‘Shunrei’ seedlings were inoculated with *F. culmorum* and incubated at 23 °C with a 14-h photoperiod. After 72 h, the root metabolites were extracted in 80% methanol for 24 h, and the obtained extract was evaporated in vacuo. The resulting residue was subjected to ODS column chromatography (Cosmosil 75C18-PREP; Nacalai Tesque, Kyoto, Japan), using 20%, 40%, 50%, 60%, 70%, and 80% methanol. Because **3** was eluted in both the 50% and 60% methanol fractions, the two fractions were combined, evaporated to dryness, and subjected to silica gel column chromatography, using mixtures of acetone and hexane (20%, 30%, 40%, 50%, and 60% acetone; 1.0 L each). Because compound **3** was detected in both the 30 and 40% acetone fractions, the two fractions were concentrated, and the resulting residue (35.8 mg) was dissolved in methanol and then subjected to preparative HPLC. Conditions were as follows: column, Cosmosil 5C_18_-AR-II 10 mm × 250 mm; solvents, water (A) and acetonitrile (B); elution, 40% B/(A+B); flow rate, 7.0 mL/min, detection, 280 nm; column temperature, 40 °C. Compound **3** was eluted at 31.1 min.

Compound **3** (*N*-cinnamoyl-(1*H*-indol-3-yl)methylamine) 2.6 mg, HR MS (positive ESI): *m/z* 277.1331 [M+H]^+^ (calcd. for C_18_H_17_ON_2_, *m/z* 277.1341); UV-Vis (acetonitrile-water containing 0.1% formic acid): λ_max_ 220 and 280 nm; ^1^H- and ^13^C-NMR data are shown in Table 1.

### 4.5. Synthesis of **3**

*N*-Hydroxysuccinimide ester of cinnamic acid was synthesized from cinnamic acid according to the methods described by Stöckigt and Zenk [30], and *N*-cinnamoyl-(1*H*-indol-3-yl)methylamine was synthesized from *N*-hydroxysuccinimide ester of cinnamic acid and IMA (Sigma-Aldrich, St. Louis, MO, USA) according to the methods described by Negrel and Smith [31]. IMA (73 mg, 1.0 mmoL) was dissolved in distilled water (10 mL), and the pH of the solution was adjusted to 8.0 using NaHCO_3_. Meanwhile, *N*-hydroxysuccinimide ester of cinnamic acid (123 mg, 0.5 mmoL) was dissolved in acetone (10 mL) and mixed with IMA solution at room temperature for 24 h. The resulting solution was then acidified using acetic acid (1.5 mL), concentrated by removing the acetone via evaporation, and then extracted three times using 20 mL ethyl acetate. The ethyl acetate layer was dried over Na_2_SO_4_ overnight and concentrated to dryness, and the resulting residue was subjected to silica gel column chromatography (Daisogel IR-60-63-210, 80 g; Osaka Soda, Osaka, Japan), using a 3:7 (*v*:*v*) mixtures of acetone and hexane and a fraction volume of 5 mL. The fractions that contained **3** were combined and concentrated.

Compound **3** (119 mg, yield 86.4%). ESI-MS: *m/z*: 277.1 [M+H]^+^. NMR data were identical with the compound isolated from barley roots.

### 4.6. Analyses of Phenylamides and Amines

Plant materials were immersed in 80% methanol for 24 h, and the resulting extracts were subjected to LC-MS/MS analysis with MRM, according to the methods described by Morimoto et al. [25] and Ube et al. [11], using the following LC conditions: column, Acquity UPLC BEH C18 column 2.1 × 50 mm (1.7 μm; Waters); flow rate, 0.2 mL/min; column temperature, 40 °C; solvents, 0.1% formic acid in water (A) and 0.1% formic acid in acetonitrile (B); gradient, 5–70% A/(A + B) within 10 min. The MRM conditions were optimized using authentic compounds.

### 4.7. BLAST Analysis and Phylogenetic Analysis

Blast search was performed using rice THT1/2 and TBT1/2 protein sequences as queries on the EnsemblPlants database (http://plants.ensembl.org/Hordeum_vulgare/Info/Index?db=core). In the blast analysis, HvTHT1-9 showed more than 65% of identities % for OsTHT1/2 or TBT1/2. The HvTHT sequences were obtained from the EnsemblPlants database, whereas the sequences of BAHD acyltransferases from other plant species were extracted from GenBank (https://www.ncbi.nlm.nih.gov/genbank/). The amino acid sequences were aligned using ClustalW, and then a dendrogram was generated using the neighbor-joining method with *p*-distance in MEGA 7 (https://www.megasoftware.net/). The robustness of the tree branches was assessed using bootstrap analysis with 1000 replicates.

Blast search was also performed using HvTHT8 protein sequences as query on GenBank and IPK Rye Blast Server (https://webblast.ipk-gatersleben.de/ryeselect/). The alignment and dendrogram were generated though the methods described above.

For detection of *HvTHT7/8* in barley genome of ‘Morex’, ‘Barke’, and ‘Bowman’, IPK Barley Blast Server (https://webblast.ipk-gatersleben.de/barley_ibsc/) was used.

### 4.8. RNA and DNA Extraction and qRT-PCR Analysis

Total RNA was extracted from barley ‘Shunrei’ tissues using ISOGEN II (Nippon Gene, Tokyo, Japan) and reverse-transcribed using the PrimeScript RT Reagent Kit with gDNA Eraser (Takara Bio, Kusatsu, Japan). Using total cDNA as a template, qRT-PCR was then performed using SYBR Green Realtime PCR Master Mix (Toyobo, Osaka, Japan) and gene-specific primers (Appendix A). The transcript levels of genes of interest were normalized to the expression of the ADP-ribosylation factor-like protein (ADP) gene [32], and relative expression (log2) was calculated by subtracting the Cq values of genes of interest from those of the ADP gene. Fold changes were calculated as 2*^log2^*, where *log2* represents relative expression.

Genomic DNA was extracted from barley ‘Shunrei’ and ‘Morex’ tissues by the modified cetyltrimethylammonium bromide method of Murray and Thompson [33] and stored at −20 °C until use.

### 4.9. Molecular Cloning of Candidate Genes

The cDNA and *p*Cold GST vectors (*pGST*, Takara Bio) were amplified using *HvTHT2*, *HvTHT8*, and linearized *pGST* primers, respectively (Appendix A), Prime STAR MAX (Takara Bio), and the following PCR conditions: initial denaturation at 98 °C for 30 s, followed by 35 cycles of 98 °C for 10 s, 60 °C for 10 s, and 72 °C for 10 or 15 s. PCR products of the desired size were gel-purified (Promega, Wisconsin, USA) and inserted into the linearized *p*Cold GST using the In-Fusion cloning kit (Takara Bio). The resulting plasmids (*HvTHT2*/*pGST* and *HvTHT8*/*pGST*) were transformed into *E. coli* DH5, and then clones were individually purified using a Miniprep kit from Qiagen (Venlo, Netherlands) and, finally, sequenced. The sequences of the inserted region in the *pGST* and the PCR products of HvTHT7/8 were verified by DNA sequencing with Applied Bio Systems 3500xl Genetic Analyzer (Thermo Fisher Scientific).

To confirm the genomic sequence of HvTHT7/8, the genomic DNA of ‘Shunrei’ and ‘Morex’ were amplified using the HvTHT7 primers and the PCR products were purified with the same method described above. The sequence of the genomic HvTHT7/8 fragment1 were sequenced.

Meanwhile, the *HvTHT7*/*pGST* plasmid was generated by site-directed mutation of the *HvTHT8*/*pGST* plasmid. Briefly, the *HvTHT8*/*pGST* plasmid was amplified using mutagenic primers (*HvTHT7*/*pGST*; Appendix A) and the following PCR conditions: 98 °C for 10 s, followed by 30 cycles of 65 °C for 10 s, and 72 °C for 30 s. The PCR product was transformed into *E. coli* DH5, and the *HvTHT7*/*pGST* plasmid was obtained though the methods described above.

### 4.10. Recombinant Protein Expression and Enzyme Assays

The *HvTHT2/pGST*, *HvTHT7/pGST*, *HvTHT8/pGST,* and *pCold-GST* plasmids were transformed into *E. coli* BL21 (Takara Bio) according to the manufacturer’s instructions. The resulting transformants were grown on LB medium (per liter: 10 g yeast extract, 10 g peptone, 5 g sodium chloride, and 15 g agar) that contained 100 ppm ampicillin and then selected using colony PCR. Positive colonies were grown in LB liquid medium with the antibiotics, incubated at 37 °C for 5–6 h (OD_600_ 0.5–0.6), maintained at 15 °C for 30 min, mixed with IPTG (final conc, 500 µM), maintained at 15 °C for 24 h, and centrifuged at 5000× *g* for 10 min. The resulting centrifugate (pelleted cells) was resuspended in extraction buffer (20 mM Na-Pi buffer pH 8.0, 10% glycerol), sonicated, centrifuged at 15,000× *g* for 10 min, and then subjected to TALON metal affinity resin immobilization (Takara Bio). The His-GST-tagged proteins were then purified according to the manufacturer’s instructions, and imidazole was removed from the purified fraction using a PD-10 column (GE Healthcare, Chicago, IL, USA).

To determine the kinetic parameters of HvTHT7 and HvTHT8, enzyme reactions (50 μL) were performed at 35 °C in 100 mM Tris-HCl buffer (pH 8.0) that contained 5 μL of diluted enzyme solution and different concentrations of acyl donor and acyl acceptor. To determine the kinetic parameters for the acyl donors, the reactions were performed using Cin-, Cou-, and Fer-CoAs (1–200 μM), with tryptamine (1 mM) as the acyl acceptor, and to determine the kinetic parameters for the acyl acceptors, the reactions were performed using tryptamine, serotonin, 8-oxotryptamine, and IMA (10–2000 μM), with Cin-CoA (200 μM) as the acyl donor. To determine the kinetic parameters of HvTHT2 for the acyl acceptors, reactions were performed using tryptamine, serotonin, and tyramine (10–2000 μM), with feruloyl-CoA (200 μM) as the acyl donor. Enzyme concentration and reaction time were adjusted to ensure that the reaction proceeded linearly, and after an incubation at 35 °C for 30 min, the reactions were stopped by adding 5 µL 6*N* HCl and 45 µL methanol. Finally, the reaction mixtures were subjected to LC-MS/MS analysis.

### 4.11. Administration of Deuterium-Labeled Compounds

Both [^2^H_5_]-CinTry and [^2^H_5_]-**2** were synthesized from [^2^H_5-_phenyl] cinnamic acid and unlabeled corresponding amines using the same procedure that was used to synthesize unlabeled compound **3**. [^2^H_5-_phenyl]-Cinnamic acid was a stock in the laboratory of Natural Product Chemistry, Faculty of Agriculture, Tottori University (Tottori, Japan). In the feeding experiments, barley roots were inoculated with *F. culmorum*, and after a 48-h incubation, the roots were immersed in 1 mM solutions (5 mL) of labeled [^2^H_5_]-CinTry and [^2^H_5_]-**2** in 50-mL conical tubes. The treated roots were incubated for 24 h under the growth conditions and extracted using methanol. The extracts were analyzed by LC-MS/MS in MRM methods (Appendix A). MRM method of deuterium labeled-phenylamides was developed according to Li et al. [34]

### 4.12. Anti-Microbial Activity Assay

The anti-microbial activities of **3** were evaluated using the methods described by Ube et al. [11]. Conidial-germination, germ tube-elongation, and growth-inhibition assays were performed using 10, 30, 100, 300, and 1000 μM solutions of **3**. Results are shown as the mean ± standard deviation. The significance of differences was estimated by analysis of variance using Tukey-Kramer tests. Findings of *p* < 0.05 were considered significant. Tukey-Kramer tests were performed by using Microsoft Excel software.

## 5. Conclusions

In the present study, 9-hydroxy-8-oxotryptamine (**1**), *N*-cinnamoyl-8-oxotryptamine (**2**), and *N*-cinnamoyl-(1*H*-indol-3-yl)methylamine (**3**) were identified in *Fusarium*-infected barley, designated as triticamides A–C, and characterized as barley phytoalexins. Triticamide C was a previously undescribed cinnamic acid amide, and five additional phenylamides were also observed to accumulate in pathogen-infected barley plants. Both HvTHT7 and HvTHT8 were determined to contribute to the biosynthesis of phenylamide phytoalexins in barley. The present study provides new examples of the phenylamide phytoalexins, which have been previously reported to occur in rice, oats, and wheat, and emphasizes the significance of phenylamide phytoalexins in the chemical arsenals of grass species.

## Figures and Tables

**Figure 1 ijms-20-05541-f001:**
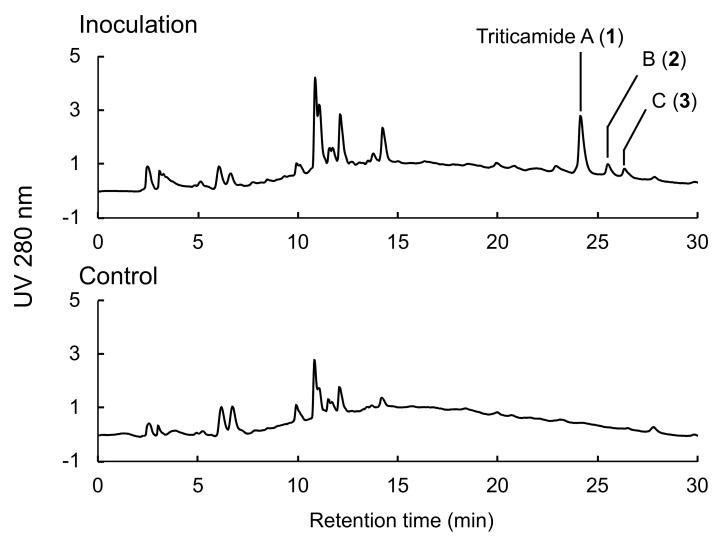
High-performance liquid chromatography (HPLC) analysis of extracts from *Fusarium culmorum*-infected and control roots. Agar plugs of *F. culmorum*-inoculated and control V8 medium were placed on the roots of 4-d-old barley seedlings, and extraction was performed at 72 h after inoculation.

**Figure 2 ijms-20-05541-f002:**
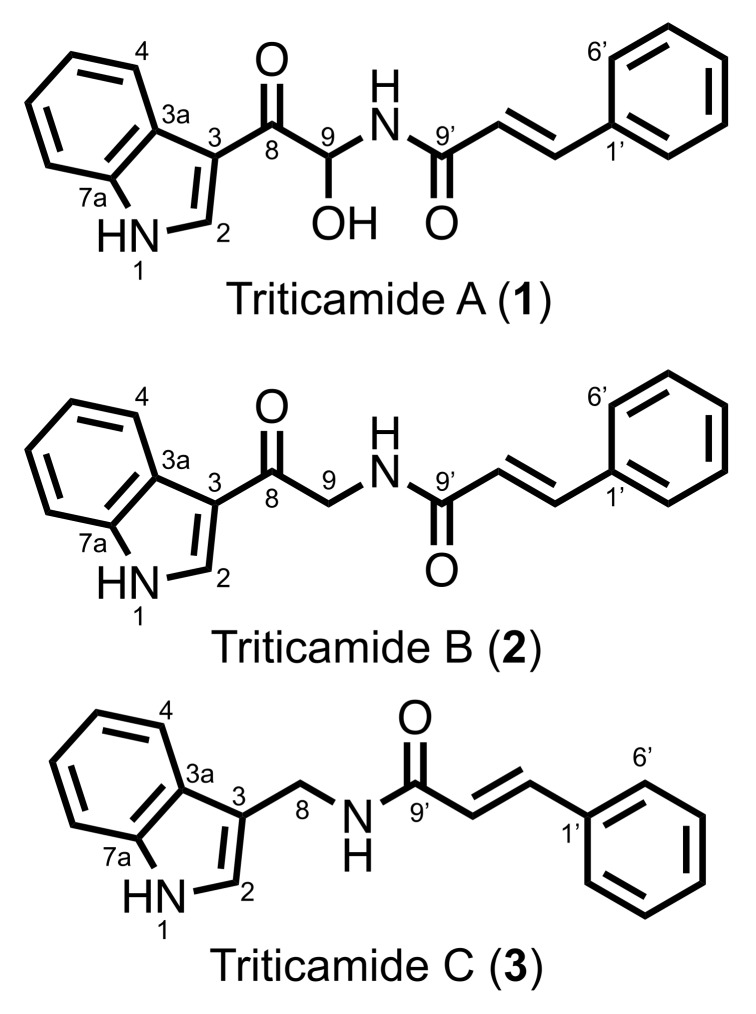
Chemical structures of **1**–**3**.

**Figure 3 ijms-20-05541-f003:**
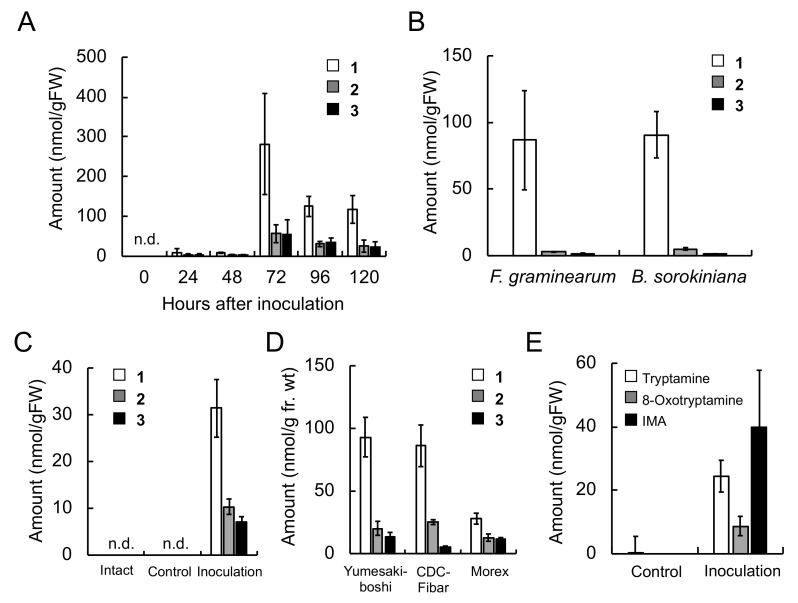
Accumulation of **1**–**3** and indole amines in pathogen-infected barley plants. (**A**) Effect of infection duration on the accumulation of **1**–**3** in *Fusarium culmorum*-infected roots. (**B**) Accumulation of **1**–**3** in *Fusarium graminearum*- and *Bipolaris sorokiniana-*infected roots at 72 h after inoculation. (**C**) Accumulation of **1**–**3** in *B. sorokiniana*-inoculated, distilled water-treated (control), and intact leaves. (**D**) Accumulation of **1**–**3** in *F. culmorum*-infected roots of barley cultivars ‘Yumesakiboshi’, ‘CDC Fibar’, and ‘Morex’ at 72 h after inoculation. (**E**) Accumulation of tryptamine, 8-oxotryptamine, and (1*H*-indol-3-yl)methylamine (IMA) in *F. culmorum*-infected roots at 72 h after inoculation. Values and error bars represent mean ± SD (*n* = 3). n.d., not detected.

**Figure 4 ijms-20-05541-f004:**
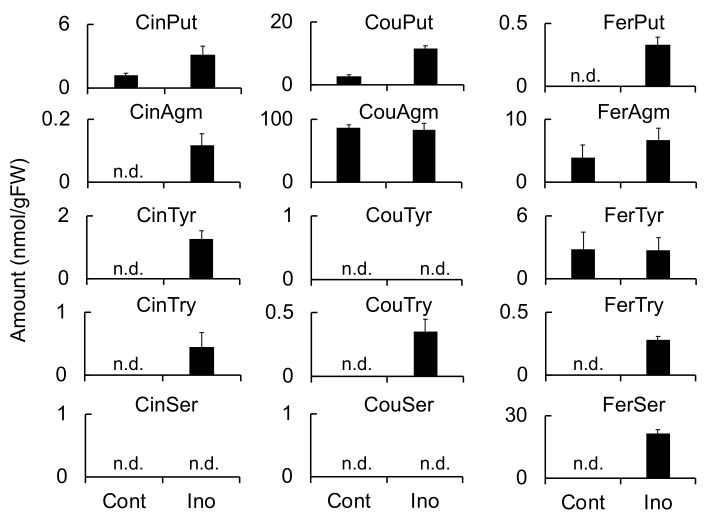
Effect of *Fusarium culmorum* infection on the accumulation of phenylamides in barley roots. Phenylamide levels were measured at 72 h after inoculation using multiple reaction monitoring (MRM) with LC-MS/MS. Values and error bars represent mean ± SD (*n* = 3). Cont, control (not inoculated); Ino, inoculated with *F. culmorum*; n.d., not detected.

**Figure 5 ijms-20-05541-f005:**
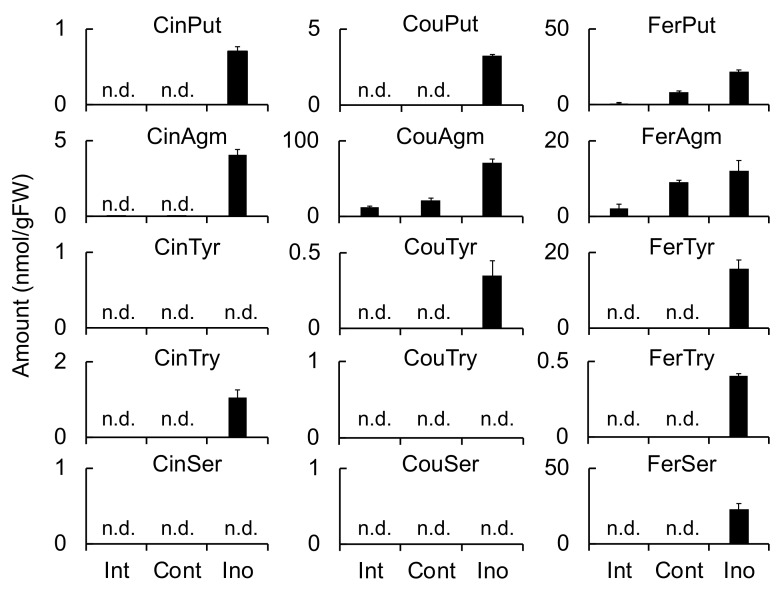
Effect of *Bipolaris sorokiniana* infection on the accumulation of phenylamides in barley leaves. Phenylamide levels were measured at 72 h after inoculation using MRM with LC-MS/MS. Values and error bars represent mean ± SD (*n* = 3). Cont, control; Ino, inoculated with *B. sorokiniana*; n.d., not detected.

**Figure 6 ijms-20-05541-f006:**
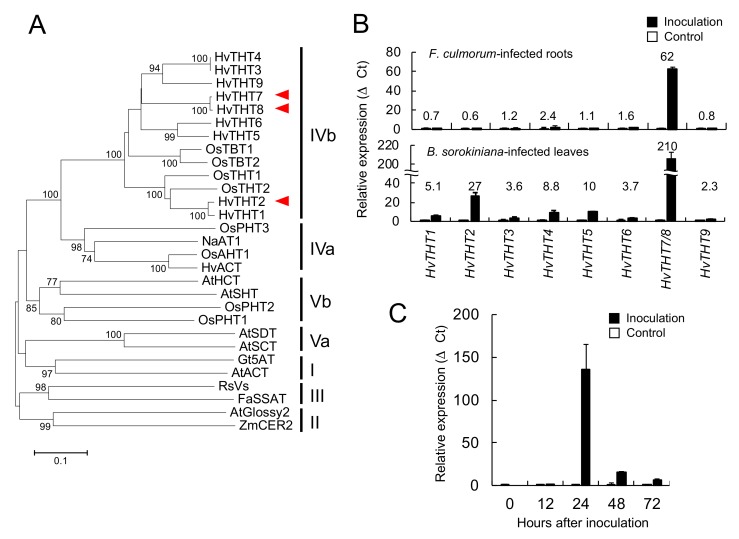
Involvement of barley tryptamine hydroxycinnamoyltransferases (HvTHTs) in the defense responses in barley. (**A**) Relationships between HvTHT proteins and BAHD-family acyltransferases from other species. A dendrogram was generated form sequences of 29 proteins in the BAHD family. Bootstrap values >70% (based on 1000 replications) are indicated at each node (bar = 0.1 amino acid substitutions per site). Non-barley sequences (species, access. no.) were obtained from GenBank: ZmGlossy2 (*Zea mays*, CAA61258), AtCER2 (*Arabidopsis thaliana*, AAM64817), FaSAAT (*Fragaria x ananassa*, AAG13130), RsVs (*Rauvolfia serpentine*, CAD89104), AtACT (*A. thaliana*, NP_200924.1), Gt5AT (*Gentiana triflora*, BAA74428), AtSCT (*A. thaliana*, Q8VZU3), AtSDT (*A. thaliana*, NP_179932), OsPHT1 (*Oryza sativa*, XP_015643300.1), OsPHT2 (*O. sativa*, XP_015641927.1), AtSHT (*A. thaliana*, NP_179497.1), AtHCT (*A. thaliana*, NP_199704.1), HvACT (*H. vulgare*, AAO73071), OsAHT1 (*O. sativa*, ANQ47369.1), NaAT (*Nicotiana attenuata*, JN390826), OsPHT3 (*O. sativa*, XP_015651503.1), OsTHT1 (*O. sativa*, XP_015613139.1), OsTHT2 (*O. sativa*, XP_015612968.1), OsTBT1 (*O. sativa*, XP_015615935.1), and OsTBT2 (*O. sativa*, XP_015615816.1). The HvTHT amino-acid sequences were obtained from the EnsemblPlants database (http://plants.ensembl.org/Hordeum_vulgare/Info/Index?db=core). Red arrowheads indicate HvTHT2, HvTHT7, and HvTHT8 that were subjected to detailed biochemical analyses. (**B**) Effect of infection by *Fusarium culmorum* and *Bipolaris sorokiniana* on the expression of *HvTHT* genes in barley roots and leaves, respectively. Total RNA was extracted 24 h after inoculation. (*C*) The kinetics of transcript levels in *F. culmorum-*infected root. In both (**B**) and (**C**), expression levels were normalized using the ADP-ribosylation factor-like protein (ADP) gene as an inner control and are expressed as relative values compared to those of control roots and leaves. Values and error bars represent mean ±SD (*n* = 3).

**Figure 7 ijms-20-05541-f007:**
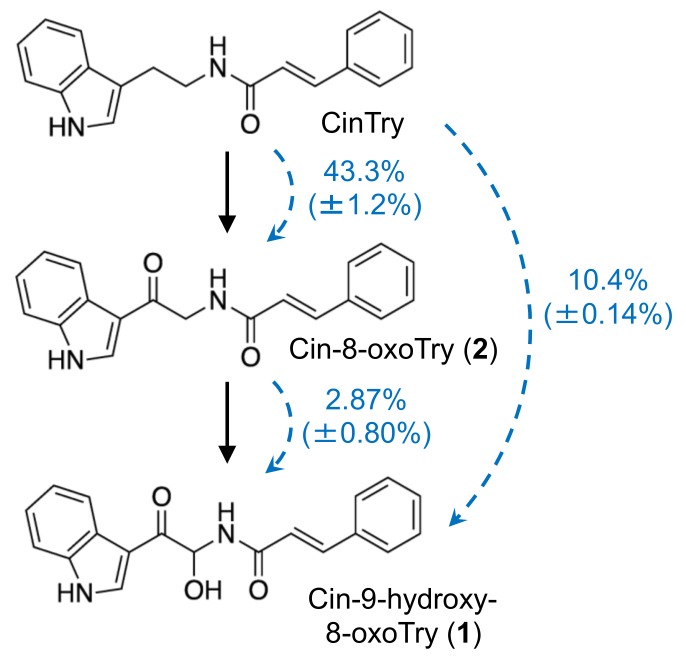
Incorporation of deuterium-labeled cinnamic acid amides into **1** and **2**. Blue arrows indicate the incorporation of labeled compounds. Percentage values indicate the mean (±SD) rates of labeled to unlabeled compound (*n* = 3). CinTry, cinnamoyltryptamine.

**Figure 8 ijms-20-05541-f008:**
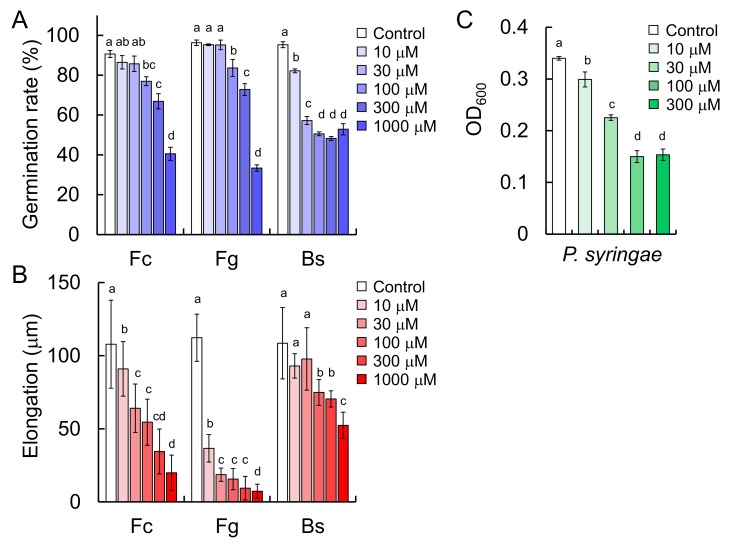
Antimicrobial activities of triticamide C. (**A**) Effect of triticamide C on the germination of *Fusarium culmorum* (Fc), *Fusarium graminearum* (Fg), and *Bipolaris sorokiniana* (Bs) conidia. Germination rates were measured at 6 h (for Fc and Fg) and 8 h (for Bs) after inoculation. (**B**) Effect of triticamide C on conidial germ tube elongation, which was measured 4 h after treatment with triticamide C. (**C**) Effect of triticamide C on the growth of *Pseudomonas syringae* pv. *japonica.* Growth was assessed by measuring OD_600_ at 0 and 24 h after the start of incubation. Values and error bars represent mean ± SD (*n* = 3). Letters (a–d) represent significant differences at *p* < 0.05 as measured using Tukey-Kramer tests.

**Table 1 ijms-20-05541-t001:** ^1^H (600 MHz) and ^13^C (150 MHz) NMR spectral data for **2** and **3** (in DMSO-*d*_6_).

*N*-Cinnamoyl-8-oxotryptamine (2)			*N*-Cinnamoyl-(1*H*-indol-3-yl)methylamine (3)		
Position	¹H muti, *J* (Hz)	¹³C	Position	¹H muti, *J* (Hz)	¹³C
NH-1	12.15 (1H, s)	-	NH-1	10.95 (1H, s)	-
2	8.57 (1H, s)	133.7	2	7.31 (1H, 2.4)	123.9
3	-	114.0	3	-	112.1
3a	-	125.4	3a	-	126.5
4	7.59 (1H, d, 7.8)	112.2	4	7.59 (1H, d, 7.2)	118.7
5	7.30 (1H, m)	121.8	5	7.10 (1H, dt, 1.2, 7.2)	121.1
6	7.32 (1H, m)	122.9	6	7.00 (1H, dt, 1.2, 7.2)	118.5
7	8.27 (1H, d, 7.2)	121.1	7	7.35 (1H, d, 7.2)	111.4
7a	-	136.4	7a	-	136.3
8	-	190.1	8	4.56 (1H, d, 5.4)	34.1
9	4.73 (2H, d, 5.4)	45.9	-	-	-
NH-10	8.53 (1H, t, 5.4)	-	NH-10	8.35 (1H, t, 5.4)	-
1′	-	134.9	1’	-	135.0
2´	7.70 (2H, d, 7.2)	127.6	2´	7.54 (2H, d, 7.2)	127.4
3´	7.47-7.58 (3H, m)	128.9	3´	7.41-7.35 (3H, m)	128.9
4´		129.5	4´		129.3
5´		128.9	5´		128.9
6´	7.70 (2H, d, 7.2)	127.6	6´	7.54 (2H, d, 7.2)	127.4
7´	7.57 (1H, d, 15.6)	138.9	7´	7.49 (1H, d, 15.6)	138.5

-: no corresponding signal.

**Table 2 ijms-20-05541-t002:** Kinetic Parameters of HvTHT7 and HvTHT8.

	HvTHT7				HvTHT8			
Substrate	*K* _m_	*k* _cat_	*k*_cat_/*K*_m_	Relative Efficiency	*K* _m_	*k* _cat_	*k*_cat_/*K*_m_	Relative Efficiency
	µM	s^−1^		%	µM	s^−1^		%
Acyl donors ^a^								
Cinnamoyl-CoA	64.2	2.06	0.032	100	62.1	2.02	0.032	100
Coumaroyl-CoA	39.2	0.67	0.017	53	35.5	0.59	0.017	52
Feruloyl-CoA	32.6	1.00	0.031	95	36.7	1.08	0.029	91
Acyl acceptors ^b^								
Tryptamine	59.2	1.58	0.027	100	60.6	1.69	0.028	100
8-Oxotryptamine	53.1	0.31	0.0059	22	54.9	0.18	0.003	12
IMA	54.2	0.37	0.0067	25	33.9	0.20	0.006	21
Serotonin	385	0.46	0.0012	5	347	0.41	0.001	4

^a^ Tryptamine (1 mM) was used as the acyl acceptor. ^b^ Cinnamoyl-CoA (200 µM) was used as the acyl donors.

**Table 3 ijms-20-05541-t003:** Kinetic parameters of HvTHT2.

Substrate	*K* _m_	*k* _cat_	*k*_cat_/*K*_m_	Relative Efficiency
	**µM**	**s** ^**−1**^		%
Acyl donors ^a^				
Cinnamoyl-CoA	25.3	0.0011	0.00004	0.028
Coumaroyl-CoA	23.6	0.0057	0.00024	0.16
Feruloyl-CoA	3.44	0.54	0.16	100
Acyl acceptors ^b^				
Tryptamine	49.7	0.0084	0.00017	100
Serotonin	49.7	0.0080	0.00016	95
Tyramine	109	0.01	0.0001	36

^a^ Tryptamine (1 mM) was used as the acyl acceptor. ^b^ Feruloyl-CoA (200 µM) was used as the acyl donor.

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
