# Peer review of "Biosynthesis of Phenylamide Phytoalexins in Pathogen-Infected Barley"

_ijms, 2019, doi:10.3390/ijms20225541_

Round 1

Reviewer 1 Report

This manuscript describes the identification and biosynthesis of phenylamide phytoalexins in barley. The article is recommended to be accepted after the revision of several minor points as follows.

- There are no Figs. 4 and 5.

- In Fig. 8, units for concentration and length are needed to be amended.

- Table 1 is better to include in Supplementary materials.

- In Line 228, ‘HvTHT8 was amplid’ should be ’HvTHT8 was amplified’.

Author Response

Reviewer 1

This manuscript describes the identification and biosynthesis of phenylamide phytoalexins in barley. The article is recommended to be accepted after the revision of several minor points as follows.

Response

Thank you for positive comments for our paper. We modified the manuscript according to suggestions except that we retained table 1 in the main manuscript.

- There are no Figs. 4 and 5.

Response

Sorry for file corruption. We checked these figures in the revised manuscript.

- In Fig. 8, units for concentration and length are needed to be amended.

Response

We amended the units for concentration and length.

- Table 1 is better to include in Supplementary materials.

Response

We retained this table in the main manuscript because the finding of compound 3 is one of main issue of this paper. In the journals for natural product chemistry, the NMR data for new compound is generally included in the article.  

- In Line 228, ‘HvTHT8 was amplid’ should be ’HvTHT8 was amplified’.

Response

Corrected accordingly (line 245).

Reviewer 2 Report

My expertise limits to the biological assays, which are well designed, the results are appropriately presented and commented.

However, although the authors state that yhe antimicrobial activities of triticamides A and B have been reported previously, it would have been nice to compare their activity with that of triticamide C.

Author Response

Reviewer 2

My expertise limits to the biological assays, which are well designed, the results are appropriately presented and commented.

Response

Thank you for positive comments for our paper. We modified the manuscript according to the suggestion.

However, although the authors state that yhe antimicrobial activities of triticamides A and B have been reported previously, it would have been nice to compare their activity with that of triticamide C.

Response

Triticamide C strongly inhibited the growth of Pseudomonas syringae in comparison with triticamides A and B.We described this in the discussion section (line 362).

Reviewer 3 Report

The manuscript under review by Ube et al. explores the inducible phenylamides of barley. Their HPLC analysis identifies 3 compounds being one of them a previously undescribed one. Following this identification they perform a brief genetic analysis of the potential genes involved in their synthesis. This is a very complete analysis that starts with the identification of the compounds and analyzes their chemical characteristics, pathway, genetics and role. Furthermore the science behind it is consistent and the results support their claims. I only have very minor comments, which are the following:

-The abstract section could be improved. It is not even mentioned in what specie the analysis was carried on until the last sentence.

-In order to improve the readability of the manuscript I think that the authors should consider adding a one-sentence summary for each of the sections of the results part. This will highlight the conclusion from each of their experiments, which sometimes is not easy to understand.

-In figure 6B It would be interesting to see the values of expression of the other HvTHT genes at 48 and 72 hrs, although it seems obvious that HvTHT7/8 are the ones with higher expression level. Together with this, how do the authors explain the 48 hr delay in the expression of the genes and the effects on the accumulation of compounds 1-3? This further explanation should be added to the discussion part.

-In Figure 6B too, it would be better to break the Y axis into values from 0 to 30 and over 200, since the expression values of HvTHT7/8 hide the changes on expression for the rest of genes.

-On Figure 8 it seems that the micro symbol (μ) has been changed to the infinity symbol (∞).

-Line 224: “in response pathogen attack” should be “ in response to pathogen attack”

-Line 359: “detabase 
” should be “database”.

-Line 361: “acquired the both 
” should be “acquired both”.

Author Response

Reviewer 3

The manuscript under review by Ube et al. explores the inducible phenylamides of barley. Their HPLC analysis identifies 3 compounds being one of them a previously undescribed one. Following this identification, they perform a brief genetic analysis of the potential genes involved in their synthesis. This is a very complete analysis that starts with the identification of the compounds and analyzes their chemical characteristics, pathway, genetics and role. Furthermore, the science behind it is consistent and the results support their claims. I only have very minor comments, which are the following:

Response

Thank you for understanding the significance of our research, and we modified manuscript according to the suggestions.

-The abstract section could be improved. It is not even mentioned in what specie the analysis was carried on until the last sentence.

Response

We mentioned ‘barley’ at the beginning of the abstract.

-In order to improve the readability of the manuscript I think that the authors should consider adding a one-sentence summary for each of the sections of the results part. This will highlight the conclusion from each of their experiments, which sometimes is not easy to understand.

Response

We added conclusion sentences to some parts of result section (Lines 142, 162, 170, 206, 240, 254, 304, 343).

-In figure 6B It would be interesting to see the values of expression of the other HvTHT genes at 48 and 72 hrs, although it seems obvious that HvTHT7/8 are the ones with higher expression level. Together with this, how do the authors explain the 48 hr delay in the expression of the genes and the effects on the accumulation of compounds 1-3? This further explanation should be added to the discussion part.

Response

We also determined the expression levels other HvTHTs at 48 and 72 h after inoculation with Fusarium culmorum. The transcripts levels of HvTHT1, HvTHT3, HvTHT4, and HvTHT6 were upregulated the at 48 h after inoculation, but their fold changes were smaller than that of HvTHT7/8. The result was described in result section (Line 237) and discussed (Line 377-385). The data was added to supplementary materials as supplementary figure S2.

-In Figure 6B too, it would be better to break the Y axis into values from 0 to 30 and over 200, since the expression values of HvTHT7/8 hide the changes on expression for the rest of genes.

Response

We modified Y axis in Fig. 6B according to the suggestion.

-On Figure 8 it seems that the micro symbol (μ) has been changed to the infinity symbol (∞).

Response

We corrected accordingly.

-Line 224: “in response pathogen attack” should be “in response to pathogen attack”.

Response

Corrected accordingly (Line 242).

-Line 359: “deta base” should be “database”.

Response

Corrected accordingly (Lines 250, 397).

-Line 361: “acquired the both” should be “acquired both”.

Response

Corrected accordingly (Line 399).